# Submillimeter Sized 2D Electrothermal Optical Fiber Scanner

**DOI:** 10.3390/s23010404

**Published:** 2022-12-30

**Authors:** Mandeep Kaur, Carlo Menon

**Affiliations:** 1MENRVA Research Group, Schools of Mechatronic Systems Engineering and Engineering Science, Simon Fraser University, Surrey, BC V3T 0A3, Canada; 2Department of Health Sciences and Technology, ETH Zürich, 8092 Zürich, Switzerland

**Keywords:** optical scanner, electrothermal actuator, cantilever beam, resonance, MEMS, imaging

## Abstract

Optical scanners are used frequently in medical imaging units to examine and diagnose cancers, assist with surgeries, and detect lesions and malignancies. The continuous growth in optics along with the use of optical fibers enables fabrication of imaging devices as small as a few millimeters in diameter. Most forward viewing endoscopic scanners contain an optical fiber acting as cantilever which is vibrated at resonance. In many cases, more than one actuating element is used to vibrate the optical fiber in two directions giving a 2D scan. In this paper, it is proposed to excite the cantilever fiber using a single actuator and scan a 2D region from its vibrating tip. An electrothermal actuator is optimized to provide a bidirectional (horizontal and vertical) displacement to the cantilever fiber placed on it. A periodic current, having a frequency equal to the resonant frequency of cantilever fiber, was passed through the actuator. The continuous expansion and contraction of the actuator enabled the free end of fiber to vibrate in a circle like pattern. A small change in the actuation frequency permitted the scanning of the area inside the circle.

## 1. Introduction

The continuous evolution of science and technology in the 21st Century enabled researchers to develop devices to assist in a variety of areas ranging from agriculture [1] to medical [2], defense and the IT sector [3]. The development of optical components provided the foundation for rapid growth of biomedical imaging techniques. Optical devices have been used frequently in medical applications to image and assist with surgical procedures as they allow direct localization of lesions and malignancies reducing the number of biopsies needed for a specific diagnostic procedure [4]. An endoscopic procedure requires the distal end of the scope to be placed close to the imageable target area through an opening such as nose, mouth, etc. Thus, the size of an endoscopic device dictates the area of the body that can be imaged using that device [5].

Seibel et al. from the Washington University have proposed the design of a scanning fiber endoscope where a single mode optical fiber actuated at its based via a tubular piezoelectric actuator delivers light to the tissue surface and a ring of multimode fibers placed at the surrounding area to collect the reflected light [6]. This type of device enables the image to be captured on a temporal basis, i.e., one pixel at a time. Thus, the image is captured on a temporal basis enabling the scope diameter to be less than 3 mm without sacrificing its resolution [6].

The key component defining the size of an optical scanner is the actuator. Various optical scanners have been developed using piezoelectric [7,8,9,10], electrostatic [11,12], electromagnetic [13,14], electrothermal [15,16], and shape memory alloy [17] actuators. Among different types of actuators, an electrothermal actuator, based on the simple Joule heating effect, provides large displacements and high output forces at a low excitation voltage [18]. Thus, a very small sized electrothermal actuator can be used to provide the desired displacement. Therefore, an electrothermal actuator was used in the proposed submillimeter sized optical scanner.

This paper elaborates on the generation of a cantilevered optical fiber excited using a single electrothermal actuator for a bidirectional scanning. The work presented in this paper is a continuation of the work presented in [19,20] where an optical fiber scanner vibrating in a single direction was proposed. The original proposed design was based on exciting an optical fiber at its base along the radial direction using an electro-thermal actuator. This actuation permitted 1D motion of the fiber in ideal cases, and the entire assembly needed to be rotated to acquire a 2D scan. However, it seemed feasible to rotate the entire assembly within micrometer sized dimensions, but there were some challenges related to this approach as it required optical, electrical, and mechanical continuity between a fixed and a rotating section of the device. 

To simplify the design of the scanner and get a bidimensional scan from its distal end, one possibility included the variation of the actuator and/or cantilever design such that the fiber tip could vibrate in the 2D plane. Such design alterations avoided the use of complex FORJ (Fiber-Optic Rotary Joints) and slip rings, and are discussed in detail below in following sections. This paper presents a plausible technique to obtain a two-dimensional imaging scan using single actuator.

The paper is organized as follows: Section 2 provides an overview of the working principle of the proposed scanner. The optimization process of a MEMS electrothermal actuator to get a high bending displacement at its free end is described in Section 3. The experimental results performed using the proposed scanner are reported in Section 4 altogether with further modifications added to the actuator design to get a nearly circular shaped pattern from the vibrating fiber tip. A complete 2D scan from the scanner is reported in Section 5. A discussion and conclusions about the proposed technology are described in Section 6 and Section 7, respectively.

## 2. Working Principle

In most optical scanners, a cylindrical shaped optical fiber represents the cantilever beam. Thus, it is hard to make changes to the cantilever itself so a complete 2D scan requires either the fixed end of the cantilever to be excited in two directions [21] or the scanning assembly needs to be rotated in a fixed tube as proposed in [19]. The last among these methods in the forward viewing optical scanners requires some challenging Fiber-Optic-Rotary-Joints (FORJs) that can allow continuity of all optical, mechanical, and electrical connections within the small dimensions. Most of the hybrid FORJs available in the market are over 3 cm in diameter. Optical scanners having dimensions smaller than the centimeter range require excitation of the cantilever scanning fiber in two directions to allow for a full 2D scan.

Among the various actuations methods used in the optical scanners, electrothermal actuation is characterized by a large displacement amplitude at a given power and can be fabricated in very small dimensions [21]. Electrothermal actuators are based on the Joule heating effect where the electric resistivity causes an increase in the temperature when the current is flowing through the actuation device.

The originally proposed optical scanner was composed of a single-mode optical fiber (SMF) with a cantilevered section at its distal tip, which was excited at resonance to obtain large deflections at the tip of the fiber. In this design, the distal end of the optical fiber was reduced in diameter to about 12 μm using either the chemical etching method [19,22] or the heating and pulling method, where a multimode fiber was heated and reduced in section which was then spliced to the SMF by heating and pushing against it [20,22]. The scanning electron microscope (SEM) image of an optical fiber acting as cantilever is shown in Figure 1a.

The actuator in the original design was made from a 25 μm thick brass foil which was cut into a desired pattern having a bridge like structure using laser micromachining. The laser cutout of the brass foil is shown in Figure 1b. The bridge structure was manually lifted perpendicularly, and the cantilever fiber was placed on it. The actuator and the optical fiber were fixed together using a thin layer of epoxy (5 min epoxy, Devcon) and SU-8 collars fabricated using soft lithography [19,20]. The upper collar structures were placed on top of the fixed part of the fiber (Figure 2b), while the bottom collar structures were placed in the groove of the actuator (Figure 2a). The collars were glued together with epoxy. The cantilever-actuator assembly (Figure 2c) was covered with the heat cure epoxy (EP-17 HTND-CCM, MasterBond, Hackensack, NJ, USA) and placed in a nitinol needle having an external diameter of about 600 μm (Figure 2d). The heat cure epoxy once cured acted as a filler for the spaces and kept the two structures together to avoid a damping motion during the vibration of the cantilever end [19]. The assembly process of the sample is summarized in Figure 2.

Like most of the cantilever-based scanners, the proposed scanner was vibrated at resonance to maximize the amount of the vibration amplitude at the tip of the scanning device for a given base excitation force/displacement. For a cylindrical shaped scanner, the first resonant or natural frequency is given by:(1)$$f_{n}=\frac{\beta }{4\pi }\sqrt{\frac{E}{\rho }}\frac{R}{L^{2}}$$
where *E*, *ρ*, *R* and *L* are the Young’s modulus, density, radius and length of the cantilever, respectively, while *β* is a coefficient depending on the resonance mode selected. The free end of the cantilever followed deflection as described by Euler-Bernoulli beam theory [23].

A Square wave signal having a frequency equal to the resonant frequency of the cantilever optical fiber was passed through the actuator. The heat generated during the on phase of the actuator due to Joule heating effect expanded the bridge structure which correspondently pushed the cantilever near its fixed base. The periodic expansion-contraction of the bridge caused the fiber tip to vibrate at resonance. The vibrating tip was imaged using an optical setup made from an off-the shelf available lenses and a camera [19]. The schematic design and the actual assembled setup are represented in Figure 3 and Figure 4, respectively, where the sample was mounted on a three degrees of freedom (DOF) stage, allowing its alignment with the optical lenses. The theoretical magnification of the optical system was 18.18×, and the actual magnification of the system was measured knowing the size of the line on a resolution target and the pixel size of the CMOS (Complementary Metal Oxide Semiconductor) camera was 18.25×.

The in-plane line scan representing the tip displacement, in the case of two different samples, taken from the front (using STC-MBE132U3V camera) is shown in Figure 5. In most cases, the tip displacement was a straight line, especially when the cantilever fiber was actuated at small excitations, i.e., using a small amount of current provided to it. As the current increased at resonant frequency, the fiber tip started to whirl, i.e., it presented an elliptical shape pattern instead of a line. The change in the fiber tip displacement as a function of the current supplied is shown in Figure 6.

The cylindrical cantilever beams excited with large base excitements at resonant frequency tended to show an unstable motion at the free end of the beam in a plane perpendicular to the beam axis due to a whirling phenomenon. Haight and King studied the cross coupling between the motions of an elastic rod [24], and Hyer further developed the frequency-amplitude equations and the stability response for a cantilevered beam [25]. The stable whirling motion at the tip of a cantilever beam can be generated by exciting a cantilever beam with an excitation frequency within a small range near the resonant frequencies. Thus, a stable elliptical motion at the tip of the cantilever can be obtained by operating within a stable whirling region. 

The ellipticity of the out of plane motion generated by whirling was very small and increased with excitation force/displacement. The motion remained primarily elliptical in the frequency range close to the first resonance mode even at a high excitation force. Wu et al. discovered that the scanning pattern of a cantilevered fiber excited in single direction can be circular at high excitation forces and frequencies in a range close to the second mode. However, the overall displacement in the second mode was very small (~1/4 of the displacement at first mode) [26]. Thus, a wide complete 2D scan of the cantilever beam using a single actuator requires modifications in the cantilever-actuator design. 

The optical scanner proposed in this paper was based on the work performed in [19,20]. The actuation force was primarily in one direction in the previous work, and a small elliptical motion by the fiber tip was present due to whirling. Modifications to the actuator design were performed to get a full 2D scan from the fiber tip, and are described in this paper.

The different types of electrothermal actuators are discussed in detail in [23]. Among these, the U-shaped or hot-and-cold arm actuators are characterized by providing an in-plane bending moment at the free end or tip of the actuator. Thus, it is possible to get actuation force in two dimensions within a plane from a single actuator using a U shaped electrothermal actuator.

The commonly used U-shaped electrothermal micro-actuators for an in-plane bidirectional motion are either made of different beam/arm lengths [27] or different cross section of the two beams [28] as shown in Figure 7a,b, respectively. The working principle of these actuators consisted of using different temperatures resulting in an asymmetric thermal expansion between the two arm structures [23]. The asymmetric thermal expansion in the first case (Figure 7a) was obtained by different beam lengths. The longer arm tended to expand more than the shorter arm due to the high average temperature and length. This asymmetric thermal expansion allowed the free end of the actuator to bend towards the shorter arm.

In the case of an electrothermal actuator characterized with a different cross section, one arm is wider than the other (Figure 7b). The wider arm (also called cold arm) is usually connected to the connection anchors with a small thinner beam section called the flexure, enabling the movement of the colder arm [29]. In another configuration, the flexure arm is directly connected to the hot thinner arm while the colder arm is connected towards one of the anchors [30]. In these actuators, the asymmetric cross section of the two beams caused different current densities in the two sections. Therefore, more heat was generated in the thinner arm due to Joule heating causing a difference in the temperature between two arms. Consequently, the different thermal expansion caused the thin arm to bend towards the cold arm generating a bending moment.

## 3. Simulation for Optimization

Steady state analytical models were developed for both asymmetric length and asymmetric cross section actuators considering the dimensions of similar polysilicon micro-actuators developed in the literature to compare the results and validate the model. An electrothermal actuator with different arm lengths (design A), considered as the reference, was the one developed by Pan and Hsu in [27], while the micro-actuator with different cross sections (design B), considered as the reference, was the one developed by Huang and Lee in [28]. The material properties and dimensional parameters of both micro-actuators are summarized in Table 1.

### 3.1. Electrothermal Analysis

Electrothermal analysis of these actuators was simplified to one-dimensional models considering the actuator to be made of long and slender arms so that heat flows primarily in the longitudinal dimension. In other words, the analysis was performed by unfolding the actuator and applying the heat transfer equation to it. Moreover, the radiation was negligible at a given dimensions of structure. Thus, the steady state mathematical equation for electrothermal analysis of these MEMS micro-actuators is simplified to 1D conduction heat transfer model:(2)$$-k_{p}wh{\left[\frac{dT}{dx}\right]}_{x}+j^{2}\rho_{r}whdx=-k_{p}wh{\left[\frac{dT}{dx}\right]}_{x+dx}+2\left(h+w\right)\beta \left(T-T_{s}\right)dx$$
with *T*, *dx*, *w*, *h*, *j*, *β*, *T_s_*, *k_p_*, *ρ_r_* being the temperature along the unfolded actuator, an element of the actuator, width of the element, thickness of the actuator, current density, convection heat transfer equation, ambient temperature, thermal conductivity, and electrical resistivity of the material, respectively.

The mathematical equations to analyze the temperature distribution along similar MEMS actuators are largely described in the literature [27,30,31,32]. A similar approach was used to find the temperature profile along the actuator for both designs A and B. The equations were solved in MATLAB, and similar models are developed in COMSOL Multiphysics. The temperature distribution in both cases is reported in Figure 8 showing the perfect matching between the analysis performed using the equations and simulation. A comparison between the temperature curves in both designs is depicted in Figure 9 showing that the average temperature difference between the two arms is higher in design B.

In microsystems, it is possible that MEMS actuators are connected to a substrate at a very small gap of microns. In such cases, the heat transfer through conduction from the actuator to substrate should be included in the electrothermal analysis, which causes the average temperature difference between two arms to be even higher in both cases [28,33]. Since the electrothermal actuator considered in the design of the scanner did not have any substrate surface close to its arms structure, such heat conduction was not considered in the analysis.

### 3.2. Mechanical Deflection Analysis

The mechanical deflection analysis of an electrothermal MEMS actuator was initiated with the calculation of the thermal expansion for each arm from the temperature distribution as shown in the previous section. The variable temperature profile along the different arm structures caused the non-homogeneous thermal expansion of the actuator resulting in a bending of the tip. The change in length for each arm section is calculated as:(3)$$\Delta L=\alpha \int_{a}^{b}\left(T\left(x\right)-T_{s}\right))dx$$
where α is the linear thermal expansion coefficient, *T_s_* is the ambient temperature, *T*(*x*) is the arm temperature in function of length *x*, *a* and *b* are the start and end values of the *x* coordinate in each arm, respectively.

For the mechanical deflection analysis, both actuators were considered as a plane frame fixed at both ends at the anchor points. It was a statistically indeterminate structure having six support reactions. Thus, the structure had a degree of indeterminacy equal to three. The analytical model was developed using the force method considering three support reactions as redundant, and then the virtual-work method was used to determine the deflection of the free end of the structure. The mechanical deflection analysis using this method is explained in detail in the literature [28,34].

Considering the dimensional parameters of the actuators to be the reference actuators in both designs reported above, the performance of actuators was analyzed as a function of the input voltage. To compare the results of the two designs, the same material properties were used in this analysis. Tip deflection curves for both designs with respect to the input current are shown in Figure 10. From the tip deflection curves, it was noted that design B generated a larger tip deflection as compared to design A for a given input voltage.

### 3.3. Optimization of Design A and Design B

An electrothermal actuator with a higher tip deflection is a desired structure so that one can get the desired motion using a lower amount of input power. The temperature difference between the two arms and the corresponding tip displacement of the free end of the actuator are directly related to the length of two arm structures. The tip deflection for design A as a function of the length ratio between the two arms is shown in Figure 11a where L_c_ is the length of cold arm, and L_h_ is the length of the hot arm. All the other dimensional and mechanical properties of the actuator were the same as its reference structure. The optimal design length of cold arm was 0.52 times the length of the hot arm to get the maximum tip displacement at given input power.

In design B, the performance also depends on the width of the cold arm structure. For a given dimensions of the hot arm, the longer the wider section of the cold arm was, the higher the heat dissipation in the cold arm will be, and the higher the tip displacement will be. However, a longer cold arm portion determined a smaller flexure arm. Consequently, the flexural rigidity of the cold arm to the bending of the flexure arm and the wider cold arm increased as well. In the case of design B placed on a substrate with a small airgap, the tip deflection as a function of length ratio between the two arms (L_c_/L_h_) is reported in Figure 11b. For a given width of the cold arm, there was an optimal length ratio of 0.96 that gave the maximum tip displacement.

Similarly, the width of cold arm played an important role in the tip displacement. A wider cold arm had a smaller average temperature in that section due to a smaller current density and higher heat dissipation. Thus, a wider cold arm showed a higher tip displacement compared to a narrower one as in Figure 11b. In case the width of two arms was the same (i.e., wc = wh), the two arms expanded at the same length, and there was zero bending moment.

Analogous to this, it was also possible to check the effect of other parameters such as length and width of the hot arm. The longer the hot arm, the higher the average temperature and the tip displacement will occur. However, the maximum of the hot arm length was limited by the fact that for very long actuators, the maximum temperature can be very high causing the local melting of the arm itself. Similarly, the higher the width or thickness of the hot arm, the smaller the current density and average temperature resulting in a small tip deflection.

### 3.4. Design C

In the previous sections, it was noted that the electrothermal actuators were based on the asymmetric thermal expansion to get bending of the free end of the actuator. This asymmetry in the temperature distribution between the two arm structures was due to different lengths of the arms in design A, and due to different widths of the two arms in design B. Design C is the combination of design A and design B, where the uneven thermal expansion of the two arm structures was obtained using both different lengths and widths of the two arms. The schematic diagram of such a structure is shown in Figure 12.

The optimal dimensions between the two arm structures, obtained in the previous section, were considered as the starting point of this analysis (i.e., length of the smaller arm was 0.52 times the length of longer arm, and in the smaller arm, the length of the wider portion was 0.96 times the overall length of that section). The temperature distribution along the structure is shown in Figure 13a. The beam deflection as a function of input voltage is shown in Figure 13b.

Analogous to previous cases, the dimension of the actuator needed to be optimized. First the performance of the design C actuator was obtained as a function of the overall length of the cold arm with respect to the hot arm length, and the portion of the wider arm inside the cold arm. From Figure 14, the maximum tip displacement obtained in the case of L_c_/L_h_ was equal to 0.79, and when the wider part was about 0.96 times the overall length of the cold arm. It can also be noted that if the wider part was longer than 0.96 times, the displacement decreased as the effect of flexural rigidity became more evident.

A comparison between the tip deflection of optimized design C with designs A and B is reported in Figure 15. It can be noticed that design C showed an increase in tip deflection of about 10% with respect to design B and over 130% compared to design A.

### 3.5. Design D

In the previous analyses, no changes were made in the hot arm structure. It can be noted that the width of the hot arm also played an important role in the overall tip deflection of the free end of the actuator. When the width of the hot arm was increased, the current density decreased and consequently, the average temperature and the thermal expansion decreased. Moreover, the flexural rigidity of the hot arm also increased with its width making it more resistant to bending. Both the decreased average temperature and increased flexural rigidity decreased the overall tip displacement in a direction perpendicular to the arm length. The mechanical analysis of these electrothermal actuators showed that the crucial part in the bending of each arm structure was the portion connected to the anchors. The tip deflection was further increased by reducing the cross section of each beam in correspondence with the anchors like the flexure portion in the cold arm. The modified structure named as design D is schematized in Figure 16.

It was considered that the width of the thinner portion near the anchors was equal to two thirds of the width of the hot arm. A smaller dimension will be more favorable. However, very thin structures will be hard to fabricate, handle, and vulnerable to melting given the use of a large current density. Thus, the thinner portion was considered as 67% of the thickness of hot arm. First, the tip deflection as a function of length of the thinner part of the hot arm was considered and shown in Figure 17. The thinner portion of the flexure arm was considered as half the length of flexure arm due to its very small value. From the analysis, the optimal length of the thinner portion of hot arm (L_1_) was 1/3 of the length of hot arm (L_h_). 

The temperature distribution of this optimized design compared to the previous designs is shown in Figure 18a, while the tip deflections of these designs is reported in Figure 18b. The average temperature of the hot arm was increased from design A to D, while the average temperature of the colder arm was decreased from design A to D. Thus, the overall temperature difference became more evident in design D and consequently the thermal expansion difference between the two arms and the overall tip displacement of the free end was higher in case D. The same can also be noticed from Figure 18b. The overall tip displacement was increased by 34% with respect to the displacement of design B working under similar conditions.

## 4. Optimized Actuator-Cantilever Setup

From the analysis performed above, design D allowed a higher temperature difference between the hot and cold arm structures. Consequently, there was a higher bending of the tip in that case. The main purpose of this analysis was to obtain a higher cantilever excitation in the horizontal direction allowing the smaller axis of the ellipse described by the fiber tip to come close to a circle.

There were a few samples prepared having the bridge shapes like that of a design D MEMS electrothermal actuator. A cantilever fiber was placed at the uplifted bridge towards the hot arm as in schematics of Figure 2. In addition, a small portion of the 12 μm fiber was attached to the second half of the distal end of the cantilever, towards the colder arm side with an epoxy. The main purpose to add a smaller fiber portion to one side of the fiber was to cause asymmetry to the vibrating fiber separating the resonant frequencies in two directions and consequently avoiding the contribution of whirling. The pattern described by the free end of the cantilever tip at 20% duty cycle and 16 V of input power is shown in Figure 19.

From Figure 19, there was an increase in the smaller diameter of the ellipse. The area inside the ellipse can be imaged by changing the input power of the actuator, i.e., a decrease in the input power allowed a smaller excitation of the cantilever and consequently a smaller ellipse. However, altogether with the excitation, the average temperature of the bridge changed with the input power and caused a continuous change in the center point of the ellipse as well. Thus, the ellipse was moving as the input power of the actuator was changed.

Another possible approach to change the shape of the ellipse and sweep the area inside it was to change the excitation frequency around the resonance value by keeping the same input power. The change in the ellipse dimensions with the excitation frequency for two different samples having different cantilever fiber length is shown in Figure 20. The blue curves show the fiber tip displacement along the vertical direction, or the major axis of the elliptical shape described by the tip, while the red curves show the ellipse’s minor axis performance with respect to the frequency. The two axes of the ellipse increase or decrease in linear fashion moving closer to or further away from the resonant frequency.

### 4.1. Design E

The design modification introduced in the last section allowed the fiber tip moved in a wider ellipse than that described by the original design. However, the minor axis of the ellipse was still much smaller than the major axis. A further modification in the actuator design was performed to resemble the pattern described by the fiber tip close to a circle.

In the analyses performed for designs A, B, C, and D, the actuators consisted of two arm structures, and the current passed through both arms to have a closed-circuit loop. Thus, the current passing through the cold arm caused the heating of that arm structure as well. It is possible to have a MEMS electrothermal actuator with more than two arm structures, where the current passes through at least two arms (called hot arms, and other arms without any current are called cold arms) [35,36,37].

The analysis performed in this paper was limited to a three-armed structure. It was also noted that the change in the flexure part of the cold arm structure in design D had a very small impact on the device performance, so it was neglected in further designs.

In a three-armed electrothermal actuator, it was possible to further increase the temperature difference between the hot and cold arm structures. It was possible to ground the two arms together as in Figure 21a, allowing the current passing through the cold arm to be divided into two equal arms, i.e., small current density in each of the colder arms. Like design B, this resulted in a smaller temperature through the cold arm. Actuators having such bridge shapes were cut using the laser cutter and are shown in Figure 21b.

In design E (Figure 21b), some samples were fabricated in combination with the asymmetrical cantilever. In this case, the asymmetric fiber portion was added towards the hot side of the actuator, as in Figure 22a, and the fiber tip followed a linear scan pattern (Figure 22b). Thus, the horizontal excitation generated by the asymmetry of the bridge canceled the out of plane motion caused by the asymmetry of the cantilever tip. In the design shown in Figure 22a, some additional fine structures were included on the side of the actuator to encourage heat dissipation. When the asymmetrical fiber was added towards the colder arm configuration, the fiber tip followed an elliptical path shown in Figure 22c.

### 4.2. Final Design

The asymmetry in the fiber tip motion in design E was smaller than that of design D. It resulted from the fact that the flexural rigidity of the part connecting the two cold arms was higher than that of design D, so this design was analyzed and optimized in further design considerations.

It was possible to overcome this limitation by modifying the design as shown in Figure 23a, where one part of the cold arm was passive, and acted just as a fin to conduct heat from the two arms. The actuator pattern cut-out following this design is shown in Figure 23b.

One of the samples fabricated following this design in combination with an asymmetrical cantilever is shown in Figure 24a. This design was excited at a 50% duty cycle still showed an elliptical pattern, which became much wider in the case of smaller duty cycle. The fiber scan at a 15% duty cycle and at 12.4 V input voltage is shown in Figure 24b.

As all the scanner prototypes are handmade, the performance of each sample was different and affected the scanning pattern as well. Some samples generated a wider ellipse having a pattern close to that of a circle, while others had a scanning pattern more like an ellipse. The repeatability of a scanning pattern also varied among the samples and the driving power levels. All the samples showed a repeatable and stable behaviour if the driving power levels were considered safe enough to avoid the plastic deformation of the bridge material, i.e., the heat generated by the Joule effect was not causing displacement of the actuator surface which was not recoverable during the cooling phase. The samples were run under safe power levels for 5–10 min, which was the typical time for an endoscopy procedure, without showing any degradation over time. At very high actuation power levels, the fiber tip started to become unstable causing the scanning pattern to drift while keeping the same frequency and input power, and the performance was not repeatable. This result was mainly due to the plastic deformation of the brass which expanded beyond its reversible limit. Using the device under these conditions for a longer period of time, increased the average length of the bridge causing the fiber to bend more and break at the fixed end at a certain point. Thus, to get a repeatable and stable scanning pattern, it was desirable to run the actuator at voltages not over 7.5 V for a 50% duty cycle, and not over 15 V for 10% duty cycle.

## 5. 2D Scan

The modified version of design E enabled the fiber tip of the scanner to move in an almost circular shaped pattern as desired, and a 2D scan was obtained using the frequency sweep as discussed earlier. 

In this preliminary setup of the final design, the light from the vibrating fiber tip was imaged using the camera. Using the function generator, the vibration frequency was manually changed using 7 V of input power and a 50% duty cycle to detect the resonant frequency providing a large ellipse/circle. For one of the tested samples, the resonant frequency was 1.346 kHz, and the generated scan pattern at 13 V and a 10% duty cycle is shown in Figure 25. Using these given actuation parameters, a frequency away from the resonant frequency was determined to be 1.220 kHz where the displacement of the fiber tip was almost zero.

Once the maximum and minimum frequency were manually determined, these values were fed into the variable frequency generation control code. In the case where the variable frequency was generated fast enough to overcome the value of the frame rate of the camera, it was possible to see the complete filled area of the scanning pattern as in Figure 26, where the major diameter of the generated ellipse was about 110 μm while the minor diameter was about 90 μm. The light intensity changed within the scan area increased from the periphery to the center of the scan area due to fact that the bent of the fiber tip increased with the tip displacement making the light out of focus at that point.

## 6. Discussion

The optical fiber scanner proposed in this paper is a follow-up to the work performed in [19,20] where the tip of the scanning fiber was moving in a single direction, and it was proposed to rotate the entire assembly to get a full 2D scan area. The original proposed design of having a linear scan from the fiber tip and rotating the whole scanner required a complex system. The actuator and cantilever designs were altered to get a circular shaped scan from the distal end of the vibrating cantilever.

The principal modification in the actuator design consisted of optimizing the asymmetry between the two legs of the bridge structure allowing its free end to move in both horizontal and vertical directions. On the other hand, the cantilever tip was also made asymmetric by adding a piece of 12 μm fiber to the distal end of fiber. It is to be noted that the position of the asymmetry in the cantilever with respect to that of the actuator played a key role on the performance of the scanner prototype. A wide circular shaped scan was obtained by adding the asymmetrical fiber portion on the upper side of cantilever fiber and moving it slightly towards the colder arm of the actuator. Once a circular pattern was obtained, the 2D scan was realized by changing the excitation frequency near the resonant frequency.

Most of the available optical scanners are based on the use of more than one actuation components to excite the cantilever fiber in two directions. The smallest optical fiber endoscope had an outer diameter of 1.2 mm [5]. An imaging device based on the proposed scanner was compared with different available optical scanners in [19]. The sub-millimeter dimensions (~600 μm in diameter) was set to overcome the limitations of currently available imaging technologies to provide an earlier detection of cancer and chronic obstructive pulmonary diseases. There are about 35,000 terminal bronchioles in the lungs having an average diameter of about 0.65 mm that can be made accessible for imaging through the proposed miniaturized endoscope, whereas the current bronchoscopes are limited to visualization of the subsegmental bronchi. Thus, the proposed imaging scanner could potentially dramatically improve the diagnostic yield of a transbronchial biopsy. Such a device would enable visualization of the smallest terminal airways for the first-time providing imaging of cancerous lesions at an earlier stage. 

## 7. Conclusions

An optical scanner having a diameter of about 600 μm presenting a 2D scan from its vibrating end using a single actuator is presented in this paper. An electrothermal actuator made from thin brass foil has an active actuating part which was optimized to provide a high bending moment and bidirectional displacement. This actuator was attached near the base of the cantilever portion of an optical fiber and provided the base excitation to the fiber. The actuating frequency was matched to the resonant frequency of the fiber to get a wide circle scan. The area inside the circle was scanned by changing the vibrating frequency near the resonance frequency. An imaging device based on such a scanner may provide images of the narrow sections of the body contributing to preliminary detection of cancers and other lesions. 

## Figures and Tables

**Figure 1 sensors-23-00404-f001:**
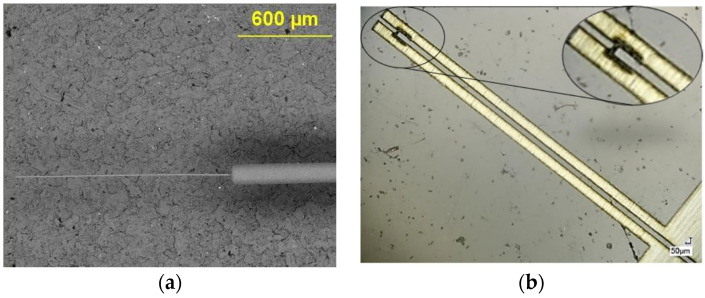
Original proposed design of the scanner: (**a**) SEM image of the cantilever fiber; (**b**) Laser cut-out of the actuator brass foil.

**Figure 2 sensors-23-00404-f002:**
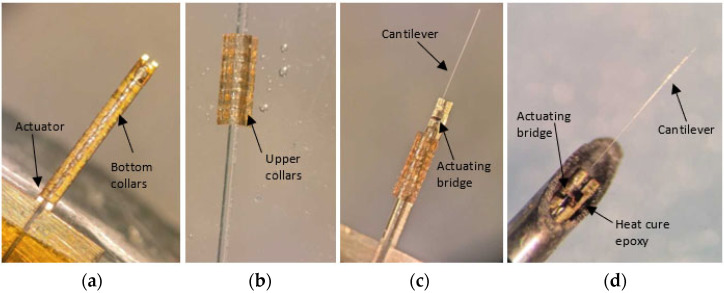
Assembly of the proposed scanner: (**a**) Bottom SU-8 collars are placed in the groove of actuator and glued together; (**b**) upper U-shaped SU-8 collars are attached to the portion of the fiber away from cantilever end; (**c**) two parts are connected; (**d**) assembly is covered with heat cure epoxy, placed in the nitinol tube and cured.

**Figure 3 sensors-23-00404-f003:**
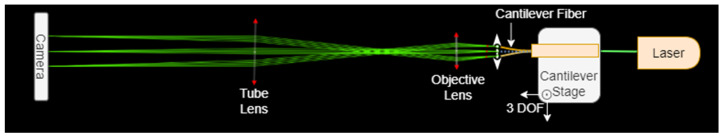
Ray diagram of the optical setup used to capture front light image.

**Figure 4 sensors-23-00404-f004:**
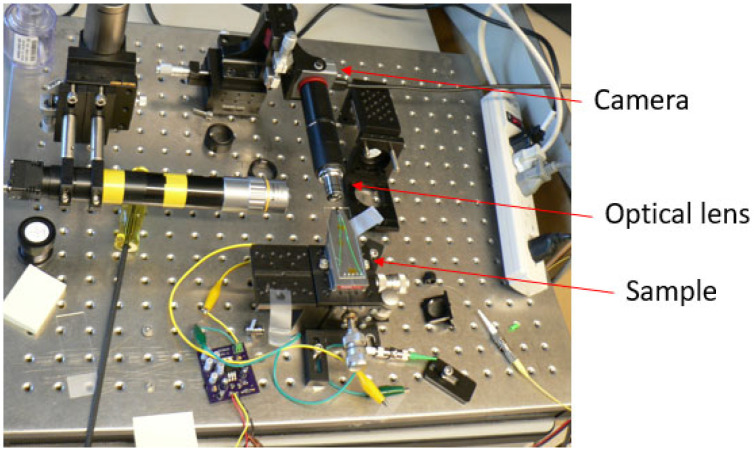
Experimental setup used to image the vibrating tip.

**Figure 5 sensors-23-00404-f005:**
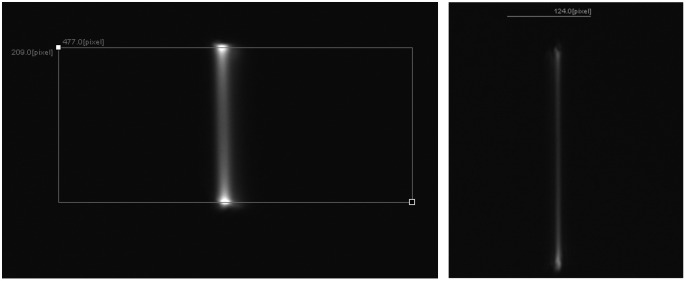
Tip displacements for two different samples using a STC-MBE132U3V camera.

**Figure 6 sensors-23-00404-f006:**
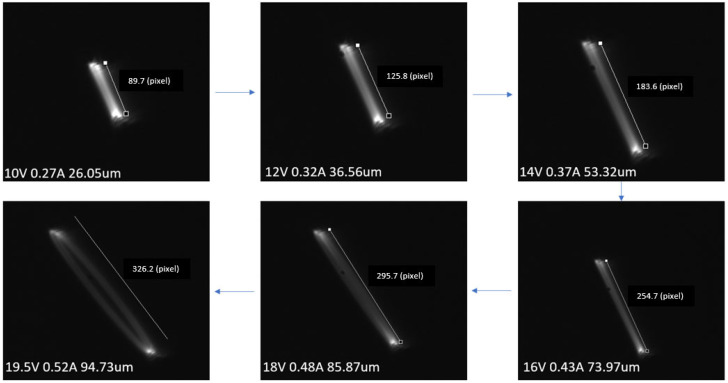
The tip displacement as a function of current.

**Figure 7 sensors-23-00404-f007:**
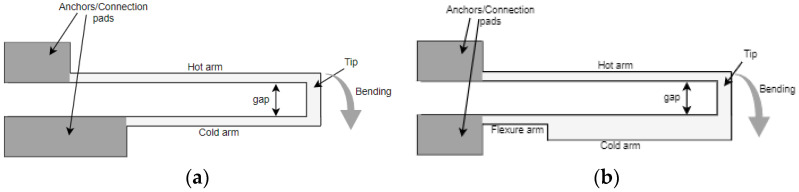
Electrothermal MEMS actuators with: (**a**) different arm lengths; (**b**) different cross section.

**Figure 8 sensors-23-00404-f008:**
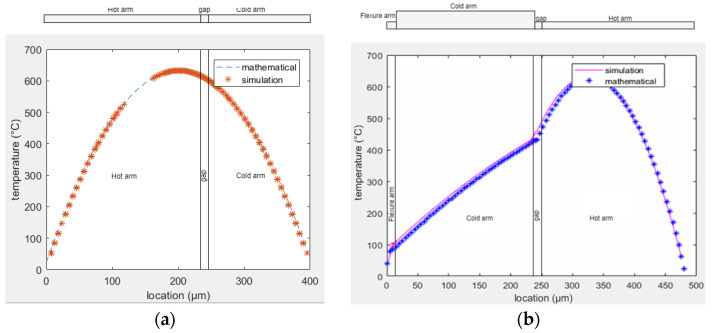
Temperature distribution along electrothermal actuators in: (**a**) design A; (**b**) design B.

**Figure 9 sensors-23-00404-f009:**
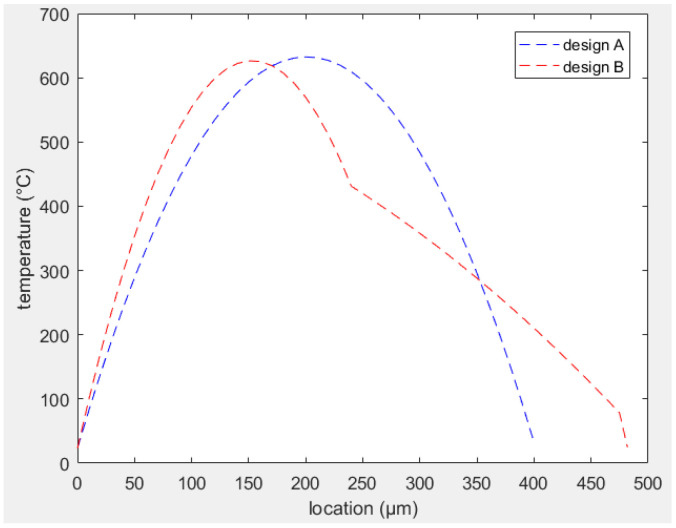
Comparison between temperature profiles in designs A and B.

**Figure 10 sensors-23-00404-f010:**
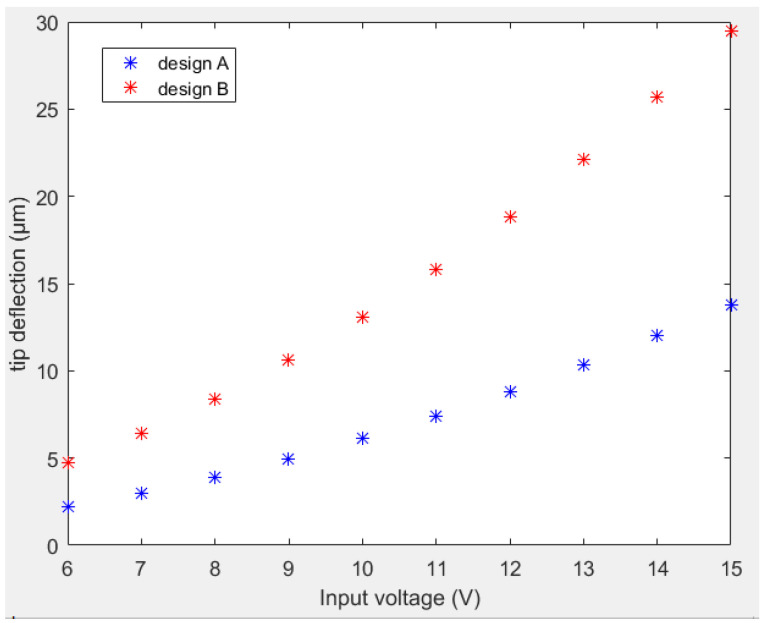
Tip deflection versus input voltage for design A (blue stars) and B (red stars).

**Figure 11 sensors-23-00404-f011:**
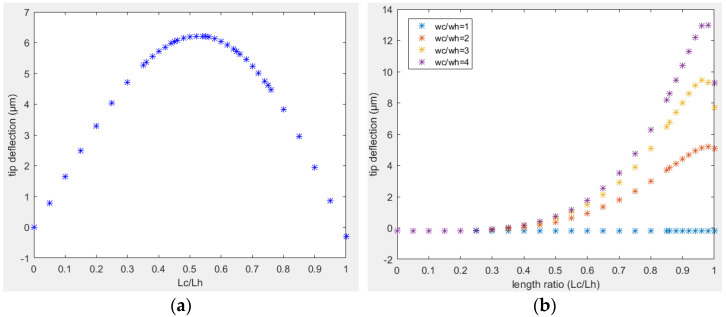
Tip deflection in function of length ratio between two arms: (**a**) design A; (**b**) design B.

**Figure 12 sensors-23-00404-f012:**
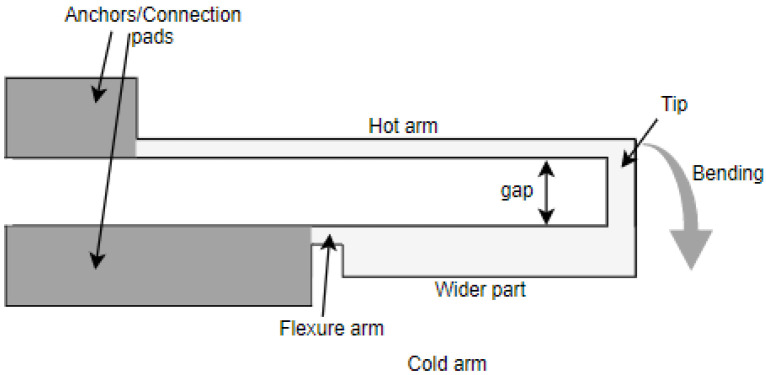
Schematic diagram of design C.

**Figure 13 sensors-23-00404-f013:**
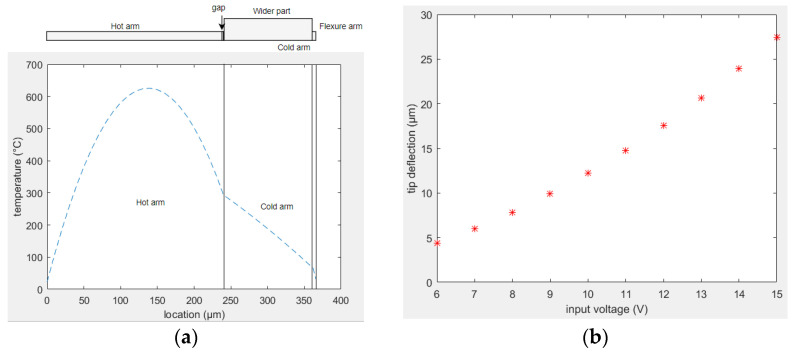
Performance of design C actuator: (**a**) temperature distribution; (**b**) tip deflection.

**Figure 14 sensors-23-00404-f014:**
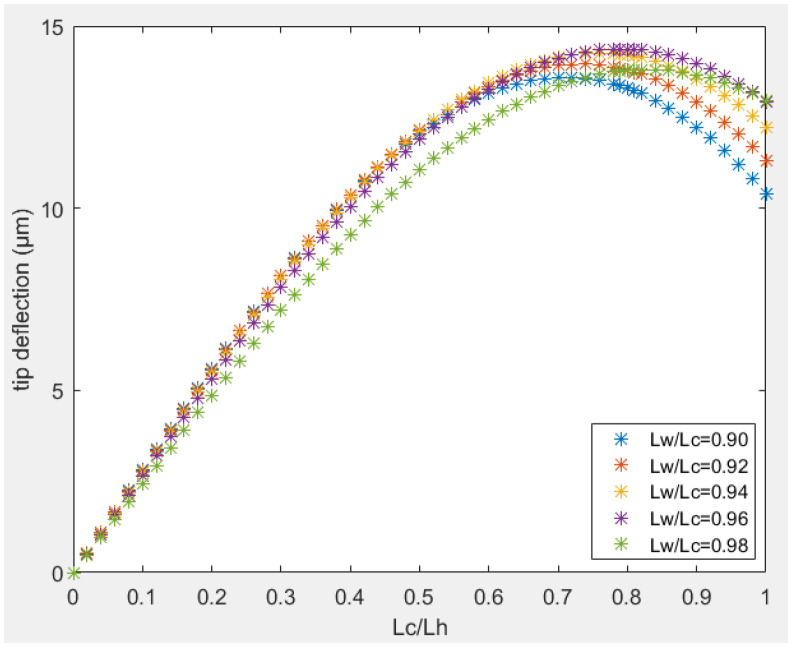
Tip deflection of design C in function of length ratio between two arm structures and the wider arm portion in the cold arm.

**Figure 15 sensors-23-00404-f015:**
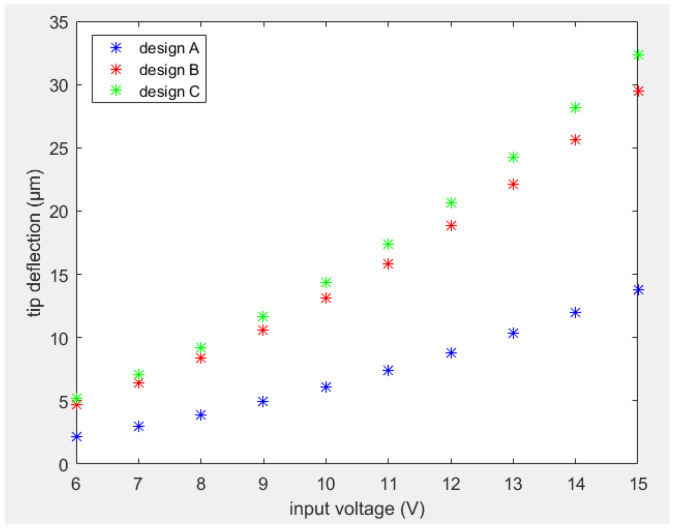
Tip deflection in function of input voltage of design C compared to designs A and B.

**Figure 16 sensors-23-00404-f016:**
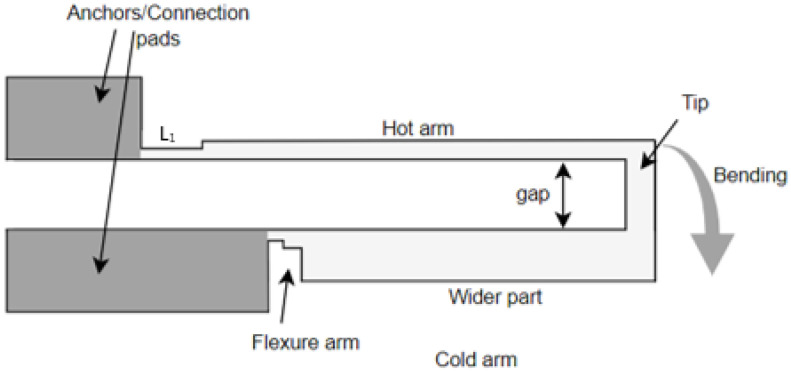
Schematic diagram of design D.

**Figure 17 sensors-23-00404-f017:**
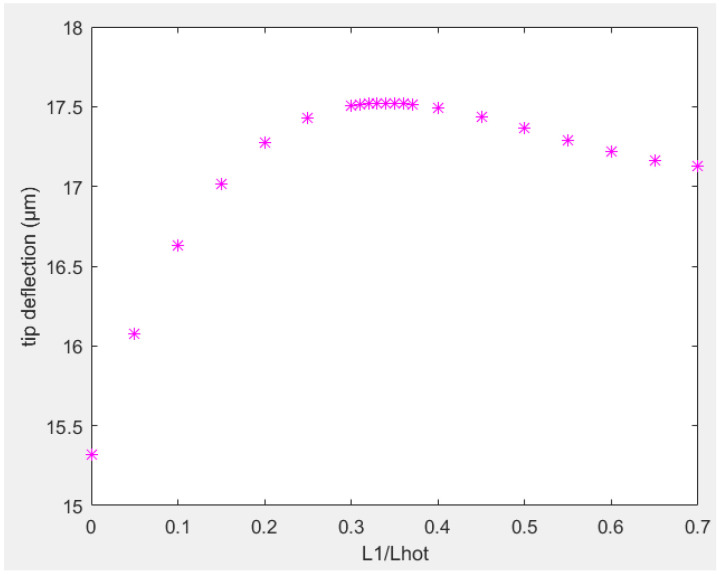
Tip deflection in function of length ratios of hot arm.

**Figure 18 sensors-23-00404-f018:**
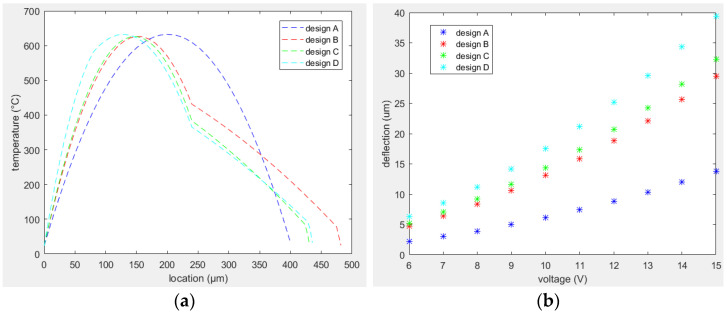
Performance of design D actuator compared to other designs: (**a**) temperature distribution; (**b**) tip deflection.

**Figure 19 sensors-23-00404-f019:**
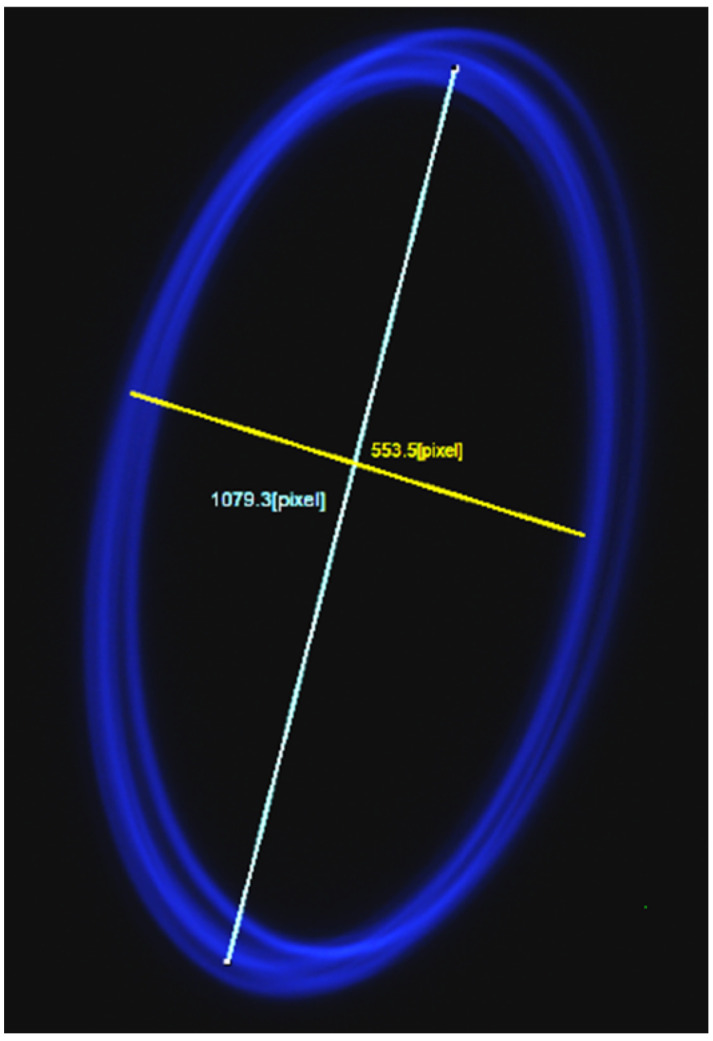
Front light image of the cantilever scan based on bridge having the shape of design D.

**Figure 20 sensors-23-00404-f020:**
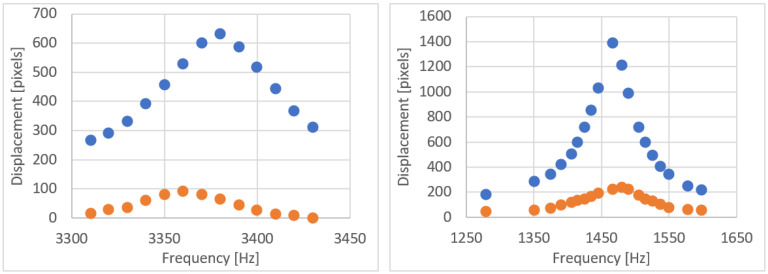
Tip displacement of the cantilever fiber (along major axis of the ellipse in blue and along minor axis of the ellipse in orange) with frequency change for two samples actuated by a bridge having shape like that of design D.

**Figure 21 sensors-23-00404-f021:**
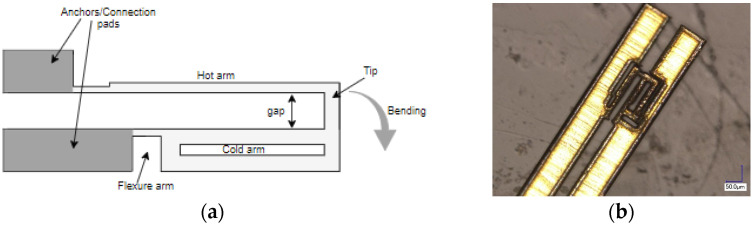
Design E having two cold arms connected at the base: (**a**) schematic diagram; (**b**) laser cut-out of the corresponding actuator.

**Figure 22 sensors-23-00404-f022:**
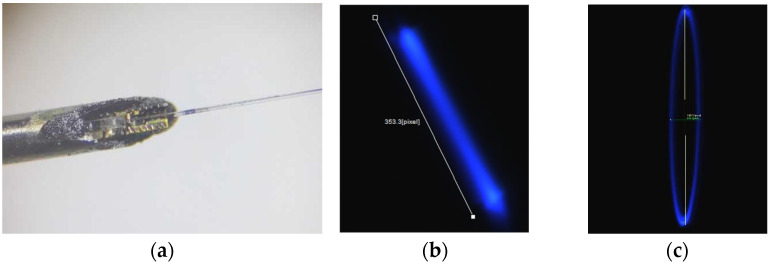
Sample with design E actuator configuration: (**a**) cantilever-actuator assembly; (**b**) scan performed by the fiber tip in case of asymmetric fiber facing hot arm; (**c**) scan from fiber tip in case asymmetry towards the cold arm.

**Figure 23 sensors-23-00404-f023:**
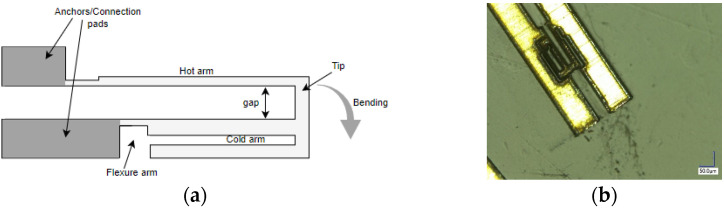
Modified version of design E having cold arm with a passive fin structure: (**a**) schematic diagram; (**b**) laser cut-out of the corresponding actuator.

**Figure 24 sensors-23-00404-f024:**
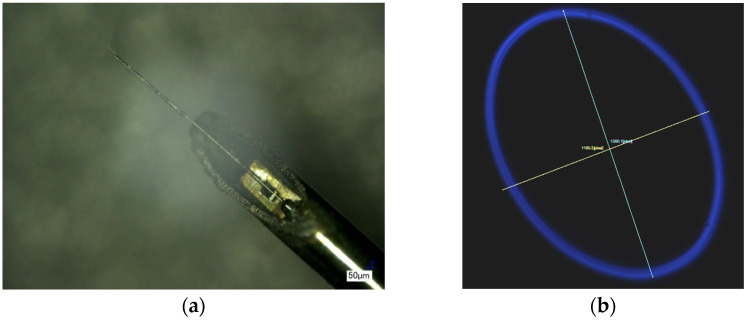
Sample with modified design E actuator configuration: (**a**) cantilever-actuator assembly; (**b**) scan from fiber tip in case asymmetry towards the cold arm.

**Figure 25 sensors-23-00404-f025:**
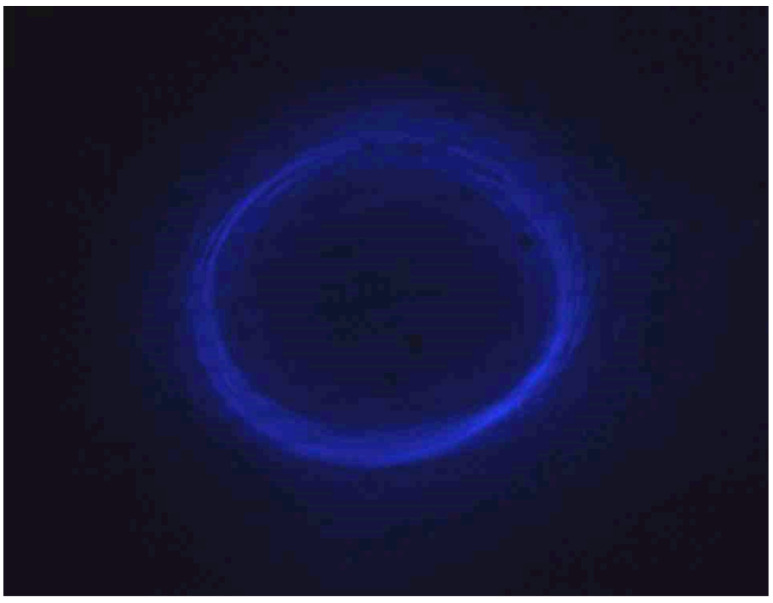
Front view of cantilever scan for vibration at its resonant frequency.

**Figure 26 sensors-23-00404-f026:**
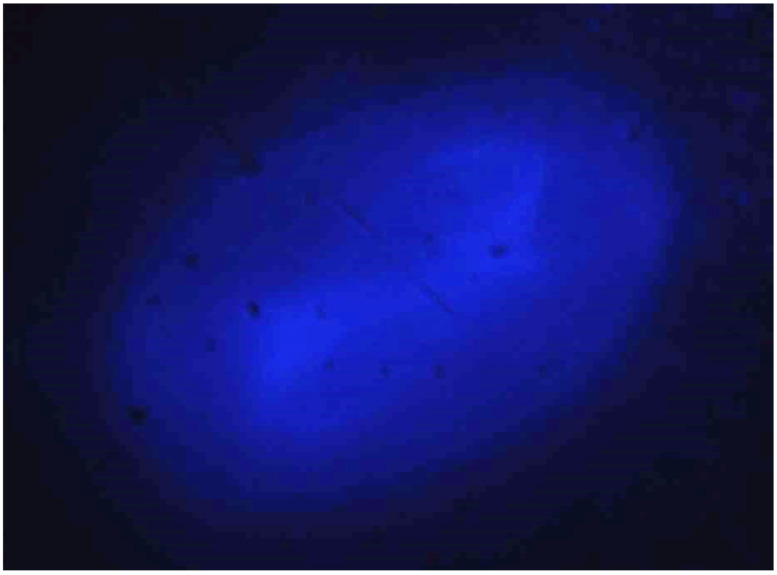
The scanning pattern from the fiber tip under the variable frequency scan.

**Table 1 sensors-23-00404-t001:** Parameters of reference polysilicon micro-actuators.

	Design A [27]	Design B [28]
**Material properties**		
Young’s modulus	150 × 10^9^ Pa	150 × 10^9^ Pa
Poisson’s ratio	0.066	0.066
Thermal conductivity	41 W/(m °C)	30 W/(m °C)
Thermal expansion coeff.	2.7 × 10^−6^ °C	2.7 × 10^−6^ °C
Resistivity	5 × 10^−4^ Ω m	1.1 × 10^−5^ Ω m
**Geometric dimensions**		
Gap between beams	≥5 μm	3 μm
Length of hot arm	500–750 μm	200 μm
Length of the cold arm	100–500 μm	162 μm
Width of beams	1–3 μm	14 μm (cold arm); 2.5 μm (hot arm)
Thickness of beams	2–3 μm	1–4 μm

## Data Availability

Data sharing is not applicable to this paper.

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
