# Peer review of "Submillimeter Sized 2D Electrothermal Optical Fiber Scanner"

_sensors, 2022, doi:10.3390/s23010404_

Round 1

Reviewer 2 Report

Comments and Suggestions for Authors

Solid analysis and description of this 2D, thermo-activated, fiber optic resonant cantilever scanning device.

However, the manuscript lacks providing more context and technical details on where and why are fiber optic based scanning imaging devices needed, as well as the other pre-existing techniques for performing 2D, resonant and non-resonant, scanning methods.  This would aid the reader, specially those not quite familiar with this R&D, to better appreciate your proposed device and technique.  Similarly, the manuscript does not elucidate enough on how does your proposed technique impact the light coupling compared to other methods.

Strongly suggested to improve your manuscript by adding some more details in these two specific areas.

Author Response

Response to Reviewer 2 Comments

Point 1: However, the manuscript lacks providing more context and technical details on where and why are fiber optic based scanning imaging devices needed, as well as the other pre-existing techniques for performing 2D, resonant and non-resonant, scanning methods.  This would aid the reader, specially those not quite familiar with this R&D, to better appreciate your proposed device and technique. 

Response 1: We thank the reviewer for their comments. The introduction and the discussion sections of the paper are amended to better describe the importance of the proposed imaging device. Other exciting techniques to get 2D scan are also listed in the manuscript as recommended and are highlighted in yellow.

Point 2: Similarly, the manuscript does not elucidate enough on how does your proposed technique impact the light coupling compared to other methods.

Response 2: The comparison of the proposed imaging device to the other devices is presented in the Discussions section as suggested.

Point 3: Strongly suggested to improve your manuscript by adding some more details in these two specific areas.

Response 3: We thank the reviewer for their valuable comments. The paper is amended as suggested and is demarcated by yellow highlight.

Reviewer 3 Report

Comments and Suggestions for Authors

A cantilever fiber with integrated electrothermal actuator in an endoscope needle of less than a millimeter in diameter is addressed for resonant optical scanning. Details on working principle, different device designs, modelling/COMSOL simulation, sample fabrication and test in a 2D scanning setup are given.

The paper is concise, I do not have suggestions for improvement.